# Using Metrics of a Mixture Effect and Nutrition from an Observational Study for Consideration towards Causal Inference

**DOI:** 10.3390/ijerph19042273

**Published:** 2022-02-17

**Authors:** Chris Gennings, Katherine Svensson, Alicja Wolk, Christian Lindh, Hannu Kiviranta, Carl-Gustaf Bornehag

**Affiliations:** 1Environmental Medicine and Public Health, Icahn School of Medicine at Mount Sinai, New York, NY 10029, USA; carl-gustaf.bornehag@kau.se; 2Department of Health Sciences, Karlstad University, 65188 Karlstad, Sweden; katherine.svensson@kau.se; 3Institute of Environmental Medicine, Karolinska Institutet, 17177 Stockholm, Sweden; alicja.wolk@ki.se; 4Department of Surgical Sciences, Uppsala University, 75237 Uppsala, Sweden; 5Division of Occupational and Environmental Medicine, Lund University, 22381 Lund, Sweden; christian.lindh@med.lu.se; 6National Institute for Health and Welfare, FI-00271 Helsinki, Finland; hannu.kiviranta@thl.fi

**Keywords:** WQS regression, endocrine disruptors, nutritional status, g-computation

## Abstract

Environmental exposures to a myriad of chemicals are associated with adverse health effects in humans, while good nutrition is associated with improved health. Single chemical in vivo and in vitro studies demonstrate causal links between the chemicals and outcomes, but such studies do not represent human exposure to environmental mixtures. One way of summarizing the effect of the joint action of chemical mixtures is through an empirically weighted index using weighted quantile sum (WQS) regression. My Nutrition Index (MNI) is a metric of overall dietary nutrition based on guideline values, including for pregnant women. Our objective is to demonstrate the use of an index as a metric for more causally linking human exposure to health outcomes using observational data. We use both a WQS index of 26 endocrine-disrupting chemicals (EDCs) and MNI using data from the SELMA pregnancy cohort to conduct causal inference using g-computation with counterfactuals for assumed either reduced prenatal EDC exposures or improved prenatal nutrition. Reducing the EDC exposure using the WQS index as a metric or improving dietary nutrition using MNI as a metric, the counterfactuals in a causal inference with one SD change indicate significant improvement in cognitive function. Evaluation of such a strategy may support decision makers for risk management of EDCs and individual choices for improving dietary nutrition.

## 1. Introduction

Evaluating the potential impact of environmental and dietary exposures on human health is challenged by using observational data. Estimation of effects from observational studies using regression methods is based on covariate-adjusted models, which may have bias due to confounders, collinearity among exposures, effect modification, and are generally considered to be association models rather than causal models [1]. Robins’ generalized methods (g methods) allow investigators to use observational data to estimate parameters that would be obtained in a perfectly randomized controlled trial [2,3,4]. Under certain assumptions, these estimates can be interpreted towards causality. g-computation is one way of estimating marginal quantities (i.e., not conditional on other covariates) of exposures using causal inference with the advantage of efficiency (i.e., small standard error) and stable estimates [5,6]. In short, g-computation is conducted using an initial flexible model of Y on exposures and covariates and constructing counterfactuals with predictions from the initial regression coefficients. Estimates and confidence intervals on marginal effects of exposure on Y are determined from Monte Carlo simulation and a bootstrap ensemble step. A g-computation analysis allows us to ask, “What would be the improvement in a human health outcome (Y) if environmental risk factors are reduced and exposure to beneficial (protective) factors are improved?”. This will be examined, with focus on reducing prenatal exposure to mixtures of environmental chemicals and improving dietary nutrition, along with the resulting improvement in children´s cognitive outcomes.

Recognizing that human exposure includes multiple environmental chemicals and dietary nutrition is based on dozens of nutrients, we propose the use of indices as metrics of exposure to use in g-computation. We use biomonitoring data and an empirically weighted index of multiple chemicals related to an important health outcome as a metric of a mixture effect (i.e., the joint action) of environmental exposures. We use a subject-specific nutrition index, My Nutrition Index (MNI; [7,8,9]), based on published recommended nutritional guidelines as a metric of overall dietary nutrition that incorporates dozens of macro- and micro-nutrients.

Our objective is to combine the estimation of a mixture effect of environmental chemicals of concern and the effect due to good nutrition as measured by MNI, with the construction of counterfactuals in g-computation, for a step towards causal inference. Using such a step, researchers can visualize “what if” exposure of the selected chemicals were reduced to a specific level as measured by the mixture index, and what the benefit would be in terms of the selected health-based outcome. On the other hand, researchers may ask what the benefit would be in terms of the selected health-based outcome with improved nutrition. We demonstrate the approach by extending the work of Tanner et al. [10], linking prenatal exposure to a mixture of chemicals to seven-year IQ by incorporating sex-specific weights, as they demonstrated a different effect for boys and girls. Using a WQS sex-stratified interaction model results in a single index with sex-specific weights and slope parameters. We also further adjust the model for nutrition using MNI. Finally, we conjecture how to use the resulting counterfactual to address issues around what would it take to reduce exposure to the specified level and what would be the score of MNI to improve the health outcome.

## 2. Methods

### 2.1. Study Population and Cognitive Assessment

Detailed recruitment and sample collection procedures have been previously described for the Swedish Environmental Longitudinal Mother and Child, Asthma and Allergy (SELMA) pregnancy cohort [10,11]. Participants provided written consent and the study was reviewed and approved by the Ethical Review Board (Uppsala, Sweden) (Dnr 2007/062 and Dnr 2017/177). Trained psychologists evaluated the cognitive function of children at 7 years of age using the Wechsler Intelligence Scale for Children, 4th edition, full scale IQ [10,12,13]. The current analysis includes 678 mother–child pairs with complete data on all exposures (EDC prenatal mixture of 26 chemicals), prenatal food frequency questionnaires, outcome (7-year IQ) and confounding variables (child sex; maternal nutrition index, energy, age at birth, pre-pregnancy weight, education level, Ravens IQ, and smoking status).

### 2.2. Collection of Prenatal Blood and Urine Samples

First-morning void urine and non-fasting blood samples were collected from mothers during their first prenatal visit (median 10-week gestation). Detailed analytical methods for all exposures are provided elsewhere [10]. Samples were stored at −20 °C (urine) and −70 °C (serum) at the Laboratory of Occupational and Environmental Medicine at Lund University, Lund, Sweden. The urine samples were analyzed by liquid chromatography tandem mass spectrometry (LC-MS/MS) to quantify 24 urinary analytes. Urinary analyte concentrations were adjusted for urine dilution by creatinine (i.e., adjusted analyte level = [analyte]/[creatinine]). Eight PFAS were quantified in serum using LC-MS/MS. Plasma was analyzed for 22 persistent organic pollutants (chlorinated or brominated) at the National Institute for Health and Welfare, Finland. Overall, 54 analytes were measured in urine, serum, and plasma. Following Tanner et al. [10], we summed analytes for DEHP and DINP using molar sums. DDT and its metabolite were also summed, as were 10 PCB congeners. We generally limited the analysis to compounds detectable in at least 75% of samples (Table A1), with the exception of summed variables.

### 2.3. My Nutrition Index (MNI) and Covariates

Data for maternal characteristics were collected at study entry for education, age, weight, and IQ using the shortened Ravens Standard Progressive Matrices [14,15,16]. Self-administered food frequency questionnaires were assessed in mid-pregnancy and linked to daily intake of nutrients, including energy levels [7]. The My Nutrition Index (MNI) was calculated using the nutrient data from food frequency questionnaires during pregnancy [7]. Children were characterized by sex, birthweight, and prematurity (<37 weeks’ gestation at birth). Although prematurity and birthweight may impact child IQ, we did not adjust for these variables since they may be mediators. We selected covariates for inclusion in the regression models using a directed acyclic graph (DAG; Figure A1).

### 2.4. Statistical Analysis

WQS regression consists of two steps: (i) estimation of a weighted index of standardized (i.e., quantiles) concentrations using a nonlinear model in an ensemble step where the final index is averaged across results from bootstrap samples of observations [17] or random subsets of components [18,19]; followed by (ii) a test for significance of the regression coefficient associated with the weighted index. The two steps are generally conducted in random splits of the data (e.g., 40% for weight estimation and 60% for hypothesis testing) and may be conducted in multiple random splits to address generalizability of the results [10,20]. The strength of the ensemble step is an increase in the number of correctly identified agents of concern compared to the use of a single nonlinear model in the estimation of the index [17]. This is analogous to forgoing parsimony for model prediction with correlated components. An example is some shrinkage methods are notorious for selecting one from a set of correlated components arbitrarily [21]. Similarly, using WQS regression in a single bootstrap sample may weight fewer components with nonnegligible weights compared to the final index averaged across all bootstrap samples. We used the gWQS R package version 3.0.4 for analysis [22].

We recently incorporated the capability of estimating the WQS index in the presence of interaction with either a continuous variable (e.g., BMI) or a categorical variable (e.g., sex) in the gWQS R package. The generalized linear model is parameterized to include a regression coefficient for the weighted index as a linear term and in the interaction term. For example, these terms may be $\beta_{0}+\beta_{1}WQS+\beta_{2}x+\beta_{12}xWQS$, when *x* is an indicator for a binary variable (e.g., sex). When *x* = 0, these terms would be $\beta_{0}+\beta_{1}WQS$, and when *x* = 1, they would be $\left(\beta_{0}+\beta_{2}\right)+\left(\beta_{1}+\beta_{12}\right)WQS$. Therefore, $\beta_{2}$ is the change in the intercept due to the binary variable, $\beta_{12}$ is the change in the slope of WQS due to the variable, and $\beta_{1}+\beta_{12}$ is the slope for the category when *x* = 1. This parameterization is for a WQS stratified interaction model.

The WQS stratified interaction model was used to estimate an empirical weighted index for the overall mixture effect of the 26 prenatal EDCs on 7-year IQ, adjusted by covariates [10]. To improve generalizability of the results, we evaluated 100 repeated holdout datasets where each training set (randomly selected 40% of the sample) was used to estimate a weighted index and each holdout dataset was used to estimate the association between the weighted index and outcome. As a sex-specific effect was noted by Tanner et al. [10], we used a WQS stratified interaction model which allowed for sex-specific weights (i.e., using 52 weights for the 26 chemicals scored into quantiles, 26 for boys and 26 for girls) and estimated in the presence of a potential interaction between the weighted index and sex. This parameterization not only allowed for sex-specific weights, but also sex-specific regression coefficients [23,24].

### 2.5. G-Computation

As is true for ordinary regression assumptions, we assume the WQS regression model is correctly specified with no unmeasured confounding, and that exposures predate the response. Further, there are primarily three assumptions, called *identifiability conditions*, for g-computation to estimate unbiased exposure effects [25]: (i) under *consistency*, well defined true exposures correspond to the measured exposures in the data; (ii) for (conditional) *exchangeability*, the probability of every exposure depends only on the covariates; and (iii) the *positivity assumption* assumes there is a nonzero probability that exposure could occur in all subgroups. In our data, exposures are measured using validated biomonitoring data from urine and serum samples. Exchangeability is addressed using covariates in the model that are linked to exposures. Based on temporality of the measurements and outcome, the exposures do not depend on IQ values. For the positivity assumption, exposure of the EDC mixtures is for all SELMA pregnant women, with detection rates to individual chemicals generally near 100% (Table A1).

Using the constructed WQSsc index (i.e., the WQS index centered and scaled), we conducted g-computation resulting in estimation of the mixture effect under counterfactuals such as exposure is at the mean WQS (i.e., WQSsc = 0) versus WQSsc values of −1 (i.e., one standard deviation below the mean exposure as measured by the WQS index). The comparison is the marginal direct mixture effect using the WQS index as a measure of change in the mixture effect of EDC exposures related to 7-year IQ. We used the *gcomp* function in the Risk Communicator R package on CRAN.

For further consideration, we conducted a similar g-computation analysis using My Nutrition Index. The counterfactual considered the direct effect of typical prenatal nutrition on 7-year IQ (i.e., as measured at the mean of MNI) compared to improved prenatal nutrition one standard deviation above the mean, i.e., improved nutrition as measured by MNI.

Finally, we compared different strategies for reducing exposure to the EDC mixtures using the WQS index as the weighted metric, where those most associated with a decline in IQ are more highly weighted. First, we decomposed the components of the WQS index into two categories: Persistent (i.e., PFAS: PFOA, PFAS, PFNA, PFDA, PFUnDA, PFHxS; and Persistent Chlorinated: HCB, Nonachlor, DDT and DDE, and the sum of PCBs) and Non-persistent (i.e., Phenols: triclosan, BPA, BPF, BFS; plasticizers: MEP, MBP, MBzP, DEHP, DINP, MHiDP, MCiNP, MOiNCH, DPHP; and other short-lived: TCP, PBA, 2OHPH). The weighted sum of each category (that together summed to 100%) was calculated per subject and displayed graphically. We used the counterfactual of reducing exposure where the WQS index is one standard deviation below the mean of the index. We considered two hypothetical scenarios: (1) What percent of the non-persistent chemicals would have to be reduced to achieve the goal of reducing exposure to one SD below the mean; (2) would completely eliminating the plasticizers be enough to achieve the goal? We calculated the revised value of the index under both scenarios for every subject and calculated a histogram of the WQS index with a reference line at the target value.

## 3. Results

Summary statistics for characterizing the mother–child pairs are provided in Table 1 Both urinary and serum matrices were used to measure 26 chemicals (41 analytes) in first-trimester maternal biological samples including from phenols, plasticizers (phthalate and non-phthalate), short-lived chemicals (organic flame retardant, OP pesticide, pyrethroid pesticide, a PAH), PFASs, and persistent chlorinated chemicals (including DDT/DDE and PCBs). Most were detected in 100% of the prenatal samples (Table A1).

We used a sex-stratified interaction WQS regression model which allowed for sex-specific weights and beta coefficients. The interaction between WQS and sex was borderline significant (Table 2; with 95% confidence interval based on the 100 holdout datasets: (−1.28, 5.13), with only 9% negative estimates). In such a parameterization, the beta associated with WQS is the slope for the reference group (boys), and the slope for the comparison group (girls) is the sum of the betas associated with the WQS and the interaction term. The weighted index was negative (estimated: −2.13) and significant for boys with 95% confidence interval (−4.27, −0.36) (Table 2). This indicates that as prenatal exposure to the 26 chemicals increases, as measured by the WQS index, there is a decline in seven-year IQ in boys. However, the slope for girls was diminished and not significant (estimated: −2.13 + 1.98 = −0.15; with 95% CI: (−2.32, 1.93)). Interpreting these beta coefficients in terms of unit changes in the WQS index (i.e., a decile-change in the WQS index) is complicated due to the infinite ways such a change could be achieved with changes in exposures to 26 chemicals; we subsequently address two scenarios in the framework of g-computation. In terms of the relationship with covariates in the same model (Table 2), there was a significant positive association between seven-year IQ and maternal IQ, maternal education, and the MNI; there was a significant negative association for maternal pre-pregnancy weight.

The distribution of estimated weights across the repeated training sets (Figure 1) identified chemicals of concern from each of the chemical groups. The chemicals with highest weights in boys were BPF, TCP, MBzP, PFOA, triclosan, MOiNCH, MEP, MCINP, PFHxS, PBA, BPA, and PFOS. In girls, the weights were generally lower, but DPHP, MEP, BPF, BPS, PFHxS, PFUnDA, MBzP, MOiNCH, OHPH, TCP, and the PCB sum had mean weights above the guideline cutoff of 1/52 = 0.019 (where 52 is the total number of weights, 26 for boys and 26 for girls).

We then used g-computation for marginal estimation of a causal link between the mixture effect from prenatal exposures as measured by the sex-stratified WQS index and 7-year IQ. Conditioning on the average weights used in the WQS index in the stratified interaction repeated holdout model, the mean IQ at seven years calculated at the average index was 98 (bootstrap 95% CI: 97, 100) for boys and was 102 (95% CI: 100, 103) for girls. In comparison, we constructed a counterfactual that prenatal exposure to the mixture as measured by the WQS index was decreased and fixed at one SD below the mean. In this counterfactual, the estimated IQ at seven years was 101 (95% CI: 99, 103) for boys and was 103 (95% CI: 101, 105) for girls. Thus, the marginal direct effect of the mixture had a mean difference in IQ of 2.2 (95% CI: 0.97, 3.4) for boys and 1.4 (95% CI: −0.13, 3.0) for girls. Such a change in the mean IQ with reduced exposure to the mixture of these EDCs could be evaluated as part of a health impact assessment; that is, is the counterfactual a meaningful goal for new regulatory management?

Suppose we set a “policy” to reduce the WQS index to one SD below its mean which is equivalent to the stratified interaction WQS index having a value less than 1.74 (Figure 2A), to reduce the adverse impact of the EDCs on important developmental markers (e.g., seven-year IQ). Recall, the WQS index is a stratified weighted index based on deciles from the population. To reduce the index using the pre-specified quantile designations with fixed weights would require the concentrations of many of the components be reduced. Of course, some of the components are environmentally persistent (i.e., the PFAS and persistent chlorinated, Figure 1) and cannot be readily reduced. In the SELMA cohort, roughly 26% of the WQS index for individuals is due to persistent compounds (Figure 2B). To reduce the WQS index below the target value without a change in the persistent chemicals, the remaining chemicals would have to be reduced to a point where their part of the index was 30% of the observed value (Figure 2C), where 99% of the subjects would have an adjusted WQS value below the suggested target of 1.74 (i.e., one SD below the mean). In comparison, we considered the scenario where the plasticizers were eliminated. The distribution of the adjusted WQS Index (Figure 2D), consisting only of the persistent chemicals, the phenols, and the other short-lived chemicals, is not an adequate adjustment as only 75% of the subjects fall below the suggested target. However, eliminating both the plasticizers and the other short-lived chemicals (Figure 2E) or the plasticizers and the phenols (Figure 2F) results in more than 95% of the subjects with values below the suggested target. These hypothetical scenarios demonstrate how the metric of prenatal exposure to 26 EDCs related to seven-year IQ, in a causal framework, can be used to suggest reduction strategies in environmental chemicals leading to potential improvements in public health.

In comparison, improving dietary nutrition is based on human behavior and choices, not necessarily regulatory policy changes. The MNI provides a metric for individuals to use to improve their dietary nutrition. We used g-computation for marginal estimation of a causal link between prenatal nutrition as measured by MNI and seven-year IQ. The mean IQ at seven years calculated at the average MNI (i.e., mean = 66.8; Table 1) was 99.9% (bootstrap 95% CI: 99, 101). In comparison, we constructed a counterfactual that prenatal nutrition improved one SD, to an MNI value of 80.8. The mean IQ increased to 101 (bootstrap 95% CI: 100, 102) which is a marginal direct effect of 1.2 points in IQ with 95% CI (0.3, 2.1). Although these results are on average nutrient intake and for a population, dietary choices are made by individuals based on food preferences and access. Use of a metric that measures how nutritious daily dietary choices are may provide a strategy for individuals to improve their nutrition.

## 4. Discussion

Our focus is on the use of counterfactuals to compare “what is” to “what if” using indices of adverse effects of EDCs and the beneficial effect of good nutrition during pregnancy on neurodevelopment (i.e., seven-year IQ). The “what is” condition is what is typical as measured at the average; here, the “what if” condition is based on a standard deviation change in the corresponding metric in the direction of improved childhood seven-year IQ. We extended the work of Tanner et al. (2020) to include a stratified interaction WQS index as the metric of the mixture effect from 26 EDC prenatal exposures. In this model, the sex-stratified interaction was borderline significant, indicating the negative association was stronger in boys than girls, while allowing for sex-specific weights in the index. We adjusted the model for nutrition using the MNI as a metric of overall nutrition which was significantly related to improved IQ. Qualitatively, the prenatal exposure to EDCs, particularly related to changes in seven-year IQ described here (adjusted for prenatal nutrition), are similar to those compounds determined to be of concern by Tanner et al. [10]. Only MCINP differ between Tanner et al., 2020 and the current study. We used the stratified interaction WQS index in a causal framework using g-computation to address counterfactuals for exposure reduction to a mixture of 26 EDCs and the MNI as a metric for prenatal nutrition in a counterfactual linked to seven-year IQ.

Other studies have focused on the “what if” question with g methods using single chemical exposure levels. For example, Garcia et al. [26] focused on air pollutant interventions as measured by NO_2_ or PM_2.5_ and childhood asthma incidence. They asked, “How would the incidence rate of asthma in our participants change if we could modify their exposure to regional NO_2_ (or PM_2.5_)?” Instead of focusing on beta coefficients relating conditional incidence rate ratios for a one-unit change in air pollution exposure, they presented a population intervention measure that estimates asthma incidence rates had exposure been, for example, no higher than 20 ppb NO_2_. Their motivation, similar to ours, is to move beyond the report of point estimates to potentially improve the translation of the study to policymakers. That is, instead of only interpreting the results of the study in terms of the significance of a beta coefficient, the g methods which use an index to represent multi-dimensional components (e.g., environmental chemicals for the WQS index and dozens of nutrients for the MNI) allow researchers and decision makers to take an additional step using the metric to provide targets for changes. We considered several scenarios for reducing the WQS index to a target level—i.e., an overall cut to all non-persistent chemicals and cuts based on chemical classes. Persistent chemicals such as PFAS (called the “forever chemicals”) should also be considered for reduction strategies; however, their remediation is often ineffective but is attracting intensive research seeking effective technologies for their removal from the environment [27,28]. The illustration indicated that a severe cut is necessary to reduce environmental exposures to the “what if” target value. A cut of 70% of the non-persistent chemicals would no doubt be difficult to achieve, and removing all plasticizers alone is not enough to reach the target value. Use of a weighted index in such counterfactuals may complement current strategies for risk management which generally focus on single chemicals, even though risk increases when mixtures are considered. Further, chemical combinations are generally considered based on convenient groupings (e.g., assuming additivity) and not based on human exposure patterns or unintentional mixtures [29].

Biomonitoring data has consistently demonstrated that human exposures to environmental chemicals are from multiple chemical classes across the lifespan [30]. Of particular concern are exposures to vulnerable populations, such as pregnant women, as environmental exposures during critical periods of human development may increase risk to adverse health and developmental outcomes (e.g., [31,32]). There remains the challenge of how best to evaluate the impact of such exposures to mixtures of chemicals that typically have complex correlation patterns. Braun et al. [33] proposed three broad questions to focus analysis of complex mixtures in epidemiological studies: (i) What are the potential health impacts of individual agents? (ii) What is the interaction among agents? (iii) What are the health effects of cumulative exposures to multiple agents? They outlined different analysis strategies depending on the research question.

Our focus herein is focusing on the third question regarding cumulative exposures to multiple agents. It is in line with the concept of a *mixture effect* from the toxicology literature where relevant environmental exposures may result in the phenomenon of “something from nothing” [34]. Environmental chemicals may be at exposures well below an effect level, but joint action of the components may produce significant effects. For example, consideration of the joint action may include a wide range of chemical classes that act along an adverse outcome pathway and are not restricted to chemicals with the same mechanism of action. Ignoring joint action of mixtures may lead to significant underestimation of risk. Thus, an important research strategy is constructing a metric for a mixture effect and identifying the contribution of individual components to the effect using human observational data.

In addition, there is a computational advantage in regression models to reduce multiple components that are highly correlated to a single metric, thereby reducing concern of multicollinearity where regression coefficients have inflated variances [35], and the reversal paradox [36] where regression coefficients have opposite signs due to correlation between components. One such strategy for estimating a metric for a mixture effect is WQS regression (e.g., [17,19,20,37]) which results in an empirically weighted index of quantiles of the mixture components. Components of concern are identified by non-negligible weights.

Another important multi-dimensional exposure source to human health is dietary nutrients. Optimal levels of many dietary nutrients (including calories, dietary fat, protein, carbohydrates, vitamins, and minerals) depend on personal characteristics (for example, age, height, weight, sex, activity level, medical conditions, behavior, and dietary choices). Therefore, the perfect diet for one person may be very different for another, meaning that there is no one-size-fits-all optimal diet. We developed the MNI, which accounts for these personal characteristics in evaluating the nutritional value of an individual’s dietary intake based on published guideline values for dozens of macro- and micronutrients and dietary components. It is a metric of how close each nutrient and dietary component is to guideline values based on the appropriateness of the response for the characteristics of the subject (e.g., age, height, weight, sex, activity level, behavior). It assigns higher scores for nutrient concentrations that fall within the published dietary guidelines’ recommended concentration range and assigns lower scores if intake for a given nutrient deviates from this optimal range (i.e., deficient or excess intake). It provides an overall index score ranging from 0 to 100, with higher scores reflecting a more nutritious diet. Thus, a perfect MNI score would be obtained if adequate intake of all nutrients is met. Our focus on nutrition is to address a potential positive public health message about the benefits of a balanced diet based on an individual’s characteristics and dietary choices. The MNI is comprised of dozens of nutrients and food components [7,9] and provides a metric for subject-specific overall nutrition. An area of ongoing inquiry is whether individuals will use a metric such as MNI to improve their dietary choices.

The strategy of using an index in g methods as a metric of exposures to multiple environmental chemicals assumes a single index in the direction of adverse effects. It may be the case that some components in a selected mixture have positive associations and others negative associations with the selected outcome [17]. For example, during some developmental periods, metals may be either nutrients (essential elements) or toxins (e.g., [38]). An advantage of estimating weights in a nonlinear model in the first step of WQS regression is that the analyst may constrain the estimation in a single direction, one at a time, thereby further improving the ill-conditioning in the data due to complex correlations. The analysis strategy may then incorporate the resulting index with positive association (i.e., a metric for a positive mixture effect) and an index with negative association (i.e., a metric for a negative mixture effect) in a final model to elucidate the response surface of the toxins and nutrients together and improve the interpretability of the data through the two indices. In the case of prenatal EDCs and cognitive development, the association is only in the adverse direction.

This approach of using both positive and negative indices in the same model is in contrast to the approach taken in quantile g-computation where the objective is to measure the overall effect of the mixture [39]. The measure of the overall effect is based on the sum of all regression coefficients due to quantiled exposures in the generalized linear model, which is then also included in a g-computation framework. This is not the same as a mixture effect where components have joint effects [40].

Prospective cohort studies, such as SELMA, are second only to randomized studies in producing unbiased resulting inference [1]. However, there are ethical concerns for randomized exposure of pregnant women to potentially toxic chemicals or poor diets, so observational data is used to study prenatal human exposures and developmental effects. Although observational in nature, the prospective cohort study design has an important advantage over other observational designs in that exposure measurements precede the development of health/developmental outcomes, allowing the temporal sequence of the relation to be more firmly established and minimizing the risk of recall bias [1]. The g methods are then useful under the identifiability conditions for unbiased counterfactual comparisons.

As with all studies, our study has limitations. Although we controlled for many important confounders and we claim that the identifiability conditions hold, there may still be residual confounders due to other variables that we were unable to collect. Under the assumptions of identifiability, we claim causality; with the potential for residual confounders, our strategy is perhaps better framed as a step towards causality. The results are limited to the exposure patterns of a Swedish pregnancy cohort with a non-diverse racial profile. Further, the impact of using a weighted index in a causal inference has not yet been fully explored.

## 5. Conclusions

In conclusion, we found that in a population-based pregnancy cohort, early prenatal exposure to a mixture of EDCs is associated with lower levels of cognitive functioning at age seven. This adverse association is stronger in boys with sex-specific chemical effects. However, there remains a nutritional positive association as measured by My Nutrition Index. We used a flexible model to support a causal inference with counterfactuals related to two indices measured during pregnancy and their impact on seven-year IQ: reduced exposure to the mixture as measured by the stratified interaction WQS index and to improved prenatal dietary nutrition as measured by MNI. Several hypothetical scenarios were considered to demonstrate how a metric of exposure could be used to suggest reduction strategies in environmental chemicals. Further evaluation of such a strategy may support decision makers for risk management of mixtures of EDCs. In addition, the My Nutrition Index provides a personalized metric of nutritional value that individuals may use to guide their dietary choices to improve their dietary habits.

## Figures and Tables

**Figure 1 ijerph-19-02273-f001:**
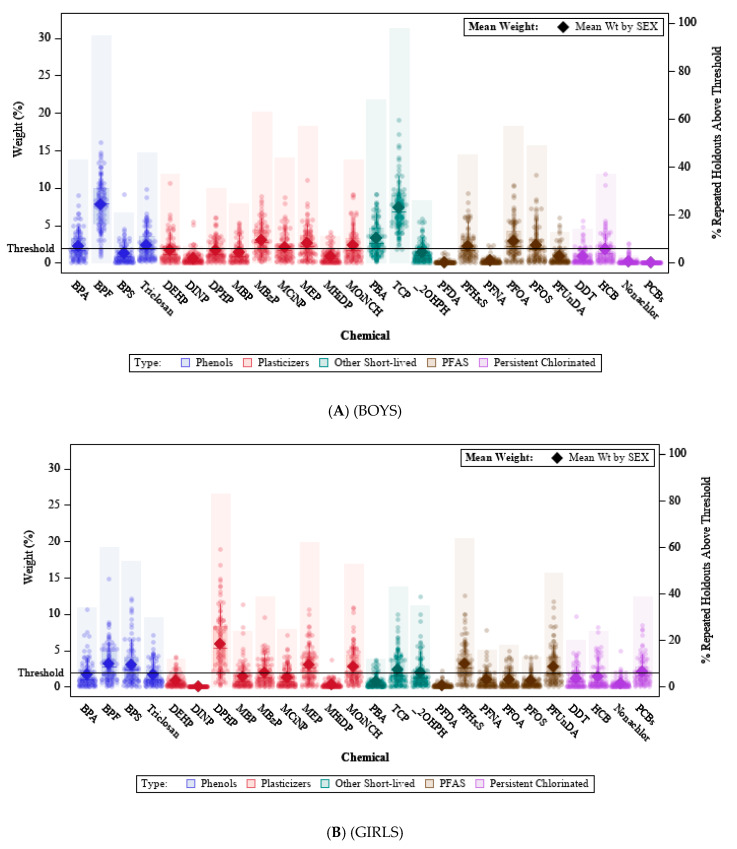
Estimated weight distribution in a WQS stratified regression model for 26 prenatal chemicals and 7-year IQ, using 100 repeated holdout validation datasets, for (**A**) boys; (**B**) girls; and (**C**) a divergent plot comparing the mean estimated sex-specific weights. Notes (**A**,**B**) Bars correspond to the right axis and indicate the percent of times a chemical exceeded the concern threshold in 100 repeated holdouts. Data points, boxplots, and diamonds correspond to the left axis. Data points indicate weights for each of the 100 holdouts. Box plots show 25th, 50th, and 75th percentiles, and whiskers show 10th and 90th percentiles of weights for the 100 holdouts. Closed diamonds show mean weights for the 100 holdouts; (**C**) The dotted lines represent the threshold guideline from the equi-weighted index (i.e., 1/(2c)), where c is the number of components.

**Figure 2 ijerph-19-02273-f002:**
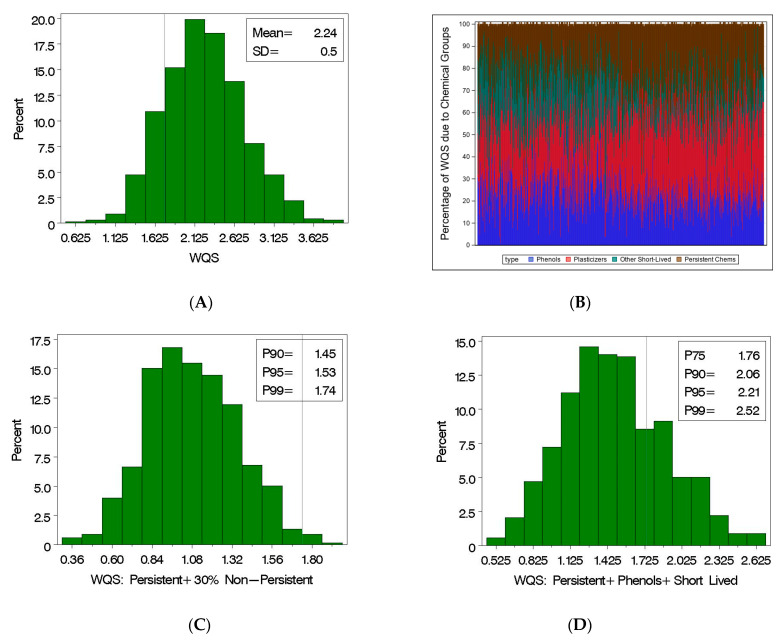
(**A**) Histogram of the stratified interaction WQS index in the SELMA cohort with reference line at selected target value (i.e., one SD below the WQS mean), (**B**) percentage of the WQS index per subject due to persistent chemicals (i.e., PFAS and persistent chlorinated), phenols, plasticizers, and other short-lived chemicals; (**C**) the distribution of the WQS index due to the persistent and 30% of the non-persistent chemicals, i.e., a 70% cut to non-persistent chemicals; the distribution of the WQS index (**D**) eliminating the plasticizers, (**E**) eliminating the plasticizers and the other short-lived compounds, and (**F**) eliminating the plasticizers and the phenols.

**Table 1 ijerph-19-02273-t001:** Summary statistics of population characteristics using the SELMA pregnancy cohort (N = 678). (*) The WQS index is derived from a WQS sex-stratified interaction model of 26 EDCs associated with child IQ at 7 years of age, adjusted by covariates.

		Mean	SD
Exposure	WQS index associated with 7-year IQ (sex-stratified, decile-scaled) *	2.24	0.50
Maternal characteristics	Graduated college n (%)	467 (69)	
	My Nutrition Index (MNI)	66.8	14.0
	Energy (kcals)	1895	545
	Age at birth (years)	31.3	4.6
	Weight in 1st trimester of pregnancy (kg)	68.8	13.5
	IQ (Raven)	114.8	14.9
	Parity	1.8	0.86
	Smoked in 1st trimester pregnancy n (%)	74 (11)	
	Creatinine (mmol/L)	10.4	4.7
Child characteristics	Female n (%)	346 (51)	
	Premature birth n (%)	25 (3.7)	
	Full Scale WISC IQ at 7 years	99.9	12.7

**Table 2 ijerph-19-02273-t002:** Parameter estimates (mean, standard error, 2.5 percentile, 97.5 percentile) from WQS sex-stratified interaction regression across 100 holdout datasets. The slope associated with WQS is for males; the interaction between WQS and sex is the difference in slopes between boys and girls.

Parameter	Estimate	Std. Error	2.5%	97.5%
(Intercept)	88.700	4.530	80.700	96.700
WQS	−2.130	1.110	−4.270	−0.359
Female	−0.622	3.190	−5.740	5.630
MNI	0.073	0.027	0.018	0.118
Energy	0.000	0.001	−0.001	0.001
Mom Age (at birth)	−0.158	0.089	−0.324	0.004
Mom Weight	−0.098	0.029	−0.156	−0.040
Mom Educ	4.790	0.862	3.090	6.470
Mom IQ	0.158	0.026	0.104	0.205
Smoker	−2.100	1.320	−4.420	0.536
WQS:Female	1.980	1.640	−1.280	5.130

## Data Availability

According to the Ethical Review Board decision and obtained personal consent, data on participating children or their mothers can not be made freely available since they constitute clinical data subject to secrecy in accordance with the Swedish Public Access to Information and Secrecy Act [OSL 2009:400]. Unique combinations of clinical data could make a study participant identifiable, and consequently, a review of secrecy may result in restrictions regarding data availability.

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
