# Peer review of "Using Metrics of a Mixture Effect and Nutrition from an Observational Study for Consideration towards Causal Inference"

_ijerph, 2022, doi:10.3390/ijerph19042273_

Round 1
Reviewer 1 Report
Thank you for giving me the opportunity to review the manuscript titled “Using Metrics of a Mixture Effect and Nutrition from an Observational Study for Consideration towards Causal Inference”.
Once the manuscript is reviewed, I add some suggestions:
- The keywords they use are not MeSH terms. Please modify your keywords and replace them with MeSH terms.
Available online: https://meshb.nlm.nih.gov/search?searchInField=termDescriptor&sort=&size=20&searchType=exactMatch&searchMethod=FullWord&q=g%20computation
- Review the citation rules for the bibliography in the text and in the references section and modify them.
- Finish the introduction section by describing the objective of your study.
- Lines 437-446: I recommend that the conclusion of your research appear in a separate section of the discussion.
Author Response
Response to Reviewer 1
We appreciate the referee reviewing our manuscript.
- We appreciate the reviewer’s suggestion about using MeSH terms. We have revised two of the key words based on the MeSH Thesaurus: endocrine disruptors and nutritional status. We, however, would like to include the two more technical terms (WQS regression and g computation) for more specific searches based on these statistical methods.
- We have revised the citation rules for the bibliography and switched to numeric references.
- We agree with the referee that the end of the introduction section should describe the objective of the study. We have included the objective in the last paragraph of the introduction section (instead of the last sentence) with a short description of the outline of the paper.
- We have highlighted the last paragraph of the Discussion section as the Conclusion, as recommended, and included additional edits to match the stated objectives.
Reviewer 2 Report
The objective of the study was to combine the estimation of a mixture effect of environmental chemicals of concern (using WQS index) and the effect due to good nutrition as measured by MNI (nutrition index), with the construction of counterfactuals in g-computation for a step towards causal inference. Overall, the manuscript is original and interesting providing a new analytical approach to assess effect of environmental mixtures as exposure, in this case EDCs, and their effect on 7-year IQ.
The authors used a DAG to guide the inclusion of potential confounders into the analysis which is also a strength of the study. There are some comments related to the DAG for the authors to consider:
- In the proposed DAG Energy seem to be a mediator between MNI and outcome and not related to prenatal EDCs so might not be included as potential confounder.
- Urinary creatinine is also included but it is only related to prenatal EDCs but is not a common cause for child IQ so it is not a confounder neither a collider. Therefore, the inclusion of creatinine as potential confounder might be not needed.
The authors used an empirically weighted index using weighted quantile sum (WQS) regression as a composite metric of EDCs prenatal exposure based on biomarkers measured in three different matrix in the context of the Swedish Environmental Longitudinal Mother and Child, Asthma and Allergy (SELMA) pregnancy cohort. The use of longitudinal data provides an important strength o the study to assess environmental exposures during pregnancy and health effects in offspring, for an exposure-outcome causal relationship that cannot be assessed in randomized clinical trials for ethical reasons. The combination of prospective data, the measurement of 26 different EDCs effect, and the use of analytical mixture assessment using an exposure index to calculate estimations and counterfactuals towards causality assessment are the main strengths of the study. The authors combine the computation of WQS index and MNI then specific targets to assess effects of counterfactual scenarios on the 7-year IQ. This approach move the statistical analysis to the usual approach of description of the state of things to the novel approach of potential effects by taking specific action and targets.
The results provided for different targets for WQS index by addressing different types of chemical restrictions, however there is not discussion related to these main results. Given that the objective of the study was using g-computation counterfactual scenarios based on targets of WQS index and MNI, more discussion is needed related to those results in terms o interpretation of the analysis. The conclusions also need to be revised based on the objetive as they are providing general conclusions of the results of EDCs in 7-year IQ but nothing related tot he objetive to this study in terms of using advanced methods to assess effects of chemical mixtures by using a WQS index and the effect of changing targets using specific counterfactuals in g-estimation.
Author Response
Response to Reviewer 2
We appreciate the referee reviewing our manuscript and providing helpful suggestions for improvement.
We appreciate the reviewer’s consideration of the DAG we presented. As typically the case, urinary concentrations should be adjusted by a dilution factor such as creatinine. Instead of including creatinine in the model, we have now adjusted each urinary concentration by the dilution factor and rerun the models. Thus, we removed creatinine from the DAG. However, energy is a bit more complicated to think about. We typically include energy in models with MNI to improve the interpretation of the nutrition index for given levels of calories. That is, energy is more of a measure of adequate food intake in contrast to MNI which is a measure of the nutritional value of the diet. Of course, both are related to the foods consumed. Willett et al (Willett et al. 1997) point out that variation in energy intake is largely due to body size and activity level so that total energy may confound associations with nutrients if these factors are related to the outcome. For this reason, we include energy in our model. But we have now changed the direction of the arrow so that MNI as a measure of nutrition may mediate the relationship of energy to the outcome.
We have added a discussion about the counterfactuals, as suggested. Largely, the illustration demonstrated that a severe cut would be necessary to achieve the target WQS value. We have also added statements in the conclusion about the counterfactuals – as suggested.
References
Willett, W. C., G. R. Howe and L. H. Kushi (1997). "Adjustment for total energy intake in epidemiologic studies." Am J Clin Nutr 65(4 Suppl): 1220S-1228S; discussion 1229S-1231S.